Longitudinal variations in the gastrointestinal microbiome of the white shrimp, Litopenaeus vannamei

Garibay-Valdez Estefanía 1
http://orcid.org/0000-0003-1116-4310 Cicala Francesco 2
http://orcid.org/0000-0003-4074-6731 Martinez-Porchas Marcel 1 marcel@ciad.mx
http://orcid.org/0000-0002-6173-4986 Gómez-Reyes Ricardo 3
http://orcid.org/0000-0002-9677-1808 Vargas-Albores Francisco 1
Gollas-Galván Teresa 1
Martínez-Córdova Luis Rafael 4
http://orcid.org/0000-0003-3502-6449 Calderón Kadiya 4
1 Tecnología de Alimentos de Origen Animal, Centro de Investigación en Alimentación y Desarrollo , Hermosillo, Sonora , México
2 Innovación Biomédica, Centro de Investigación Científica y de Educación Superior de Ensenada , Ensenada, Baja California , México
3 Universidad Autónoma de Baja California , Ensenada, Baja California , Mexico
4 Departamento de Investigaciones Científicas y Tecnológicas de la Universidad de Sonora, Universidad de Sonora , Hermosillo, Sonora , Mexico
Rahman Mohammad Shamsur
Electronic publication date: 2021 Aug 2
Publication date: 2021
Volume: 9
Electronic Location ID: e11827
Received 2021 Apr 6; Accepted 2021 Jun 30
Copyright: © 2021 Garibay-Valdez et al.
Copyright year: 2021
Copyright holder: Garibay-Valdez et al.
License: This is an open access article distributed under the terms of the Creative Commons Attribution License, which permits unrestricted use, distribution, reproduction and adaptation in any medium and for any purpose provided that it is properly attributed. For attribution, the original author(s), title, publication source (PeerJ) and either DOI or URL of the article must be cited.
License URL: https://creativecommons.org/licenses/by/4.0/

Keywords: Gastrointestinal microbiota, Crustacean gut, Gut microbiota, Gut fractions, Foregut microbiota, Hindgut microbiota, Midgut microbiota, Intestinal microbes, Crustacean microbiome

Funding: National Council for Science and Technology of México 222722 This work was supported by the National Council for Science and Technology of México (Project 222722). The funders had no role in study design, data collection and analysis, decision to publish, or preparation of the manuscript.

==============================
The shrimp gut is a long digestive structure that includes the Foregut (stomach), Midgut (hepatopancreas) and Hindgut (intestine). Each component has different structural, immunity and digestion roles. Given these three gut digestive tract components’ significance, we examined the bacterial compositions of the Foregut, Hindgut, and Midgut digestive fractions. Those bacterial communities’ structures were evaluated by sequencing the V3 hypervariable region of the 16S rRNA gene, while the functions were predicted by PICRUSt2 bioinformatics workflow. Also, to avoid contamination with environmental bacteria, shrimp were maintained under strictly controlled conditions. The pairwise differential abundance analysis revealed differences among digestive tract fractions. The families Rhodobacteraceae and Rubritalaceae registered higher abundances in the Foregut fraction, while in the Midgut, the families with a higher proportion were Aeromonadaceae, Beijerinckiaceae and Propionibacteriaceae. Finally, the Cellulomonadaceae family resulted in a higher proportion in the Hindgut. Regarding the predicted functions, amino acid and carbohydrate metabolism pathways were the primary functions registered for Foregut microbiota; conversely, pathways associated with the metabolism of lipids, terpenoids and polyketides, were detected in the Midgut fraction. In the Hindgut, pathways like the metabolism of cofactors and vitamins along with energy metabolism were enriched. Structural changes were followed by significant alterations in functional capabilities, suggesting that each fraction’s bacteria communities may carry out specific metabolic functions. Results indicate that white shrimp’s gut microbiota is widely related to the fraction analyzed across the digestive tract. Overall, our results suggest a role for the dominant bacteria in each digestive tract fraction, contributing with a novel insight into the bacterial community.

Introduction

The white shrimp, Litopenaeus vannamei, is a penaeid shrimp species with a high economic value worldwide. Despite the great production of this species, there are many problems to solve to boost the shrimp aquaculture industry. In particular, shrimp diseases lead to significant economic losses. The dysbiosis of the intestine can contribute to several shrimp diseases, such as acute hepatopancreatic necrosis disease (AHPND) and white feces syndrome (WFS) (Hossain, Dai & Qiu, 2021; Huang et al., 2020a). Recently, Wu et al. (2021) revealed that dysbiosis in the hepatopancreatic microbiota of the crayfish (Procambarus clarkii) is associated with disease outbreaks.

The white shrimp interacts with a wide variety of microbes colonizing the internal and external body (Das, Ward & Burke, 2008). The microbial communities thriving in the digestive tract can play crucial roles in the animal’s digestion, nutrition and immune response (Sommer & Bäckhed, 2013). For example, the digestive tract microbiota provides a considerable enzymatic capability, leading to manage most aspects of host physiology. Also, some tissue-specific pathways regulate the homeostatic relationship with microbiota (Belkaid & Hand, 2014).

In shrimp, the gut microbiota is implicated in health and disease (Zheng et al., 2017), optimal growth and developmental stages (Garibay-Valdez et al., 2020). Gut microbiota provides the host with beneficial functions. For instance, the microbiotas main roles include nutrient absorption, digestive enzyme production, generation of essential elements for the host’s metabolism and immune response by activating mechanisms of protection against pathogens, and the competitive exclusion of pathogenic bacteria (Gómez & Balcázar, 2008; Rowland et al., 2018; Tzuc et al., 2014). Besides, dysbiosis in the host intestinal microbiota is associated with shrimp diseases as white feces syndrome (WFS) rather than a single pathogen (Huang et al., 2020b).

Although previous studies assessed the bacterial composition in the shrimp intestine, few studies address the structural and functional changes of the microbiota among the different shrimp’s digestive tract fractions. However, recent studies have emphasized the diversity and role of the digestive tract microbiota in terrestrial (insects and mammals) and aquatic animals (fish) according to the digestive tract fraction (Brune & Dietrich, 2015; Jandhyala et al., 2015). The digestive tract fraction supports different microbial communities that are driven by distinct metabolic processes (Nielsen et al., 2017). The shrimp gut is a long digestive structure that includes the Foregut (stomach), Midgut (hepatopancreas) and Hindgut (intestine), being the intestine, the most studied fraction (Cornejo-Granados et al., 2018; Gao et al., 2019; Li et al., 2019). Each component has different structural and digestive roles, including absorption, digestion and excretion. Also, the hepatopancreas (Midgut gland) is a vital organ responsible for digestion, absorption and storage of nutrients in crustaceans, and it also plays a crucial role in regulating host innate immunity (Cornejo-Granados et al., 2017; Cheung et al., 2015; Gao et al., 2019; Rőszer, 2014). Understanding the entire shrimp digestive tract microbiota leads to a better comprehension of host-microbiota interactions and their regulatory mechanisms linked to the examined fraction from the digestive tract.

The Midgut or hepatopancreas is also a habitat of numerous bacteria that could provide specific shrimp development and physiological functions. There are only a few studies about the hepatopancreatic microbiota. For instance, Cornejo-Granados et al. (2017) described the differences in bacterial diversity in shrimp intestine and hepatopancreas; results revealed Vibrio shilonii, Faecalibacterium prausnitzii and Aeromonas taiwanensis in the intestine. However, Pantoea agglomerans and Photobacterium angustum were detected in the hepatopancreas, and the differences were associated with the digestive tract fraction (hepatopancreas or intestine) and the environment type (wild or cultured). Cheung et al. (2015) evaluated the bacterial communities from the intestine (Midgut and Hindgut), Foregut (stomach) and hepatopancreas of the cherry shrimp Neocaridina denticulata. Results revealed distinct bacterial composition in the hepatopancreas concerning the gut during the gonadal development. The mycobiota in the shrimp digestive tract was evaluated, registering a high diversity and a taxonomic correspondence to the type of organ (gut or hepatopancreas) (Li et al., 2019). Tzuc et al. (2014) isolated Vibrio strains from the shrimp stomach, hepatopancreas, and intestine (L. vannamei). These strains showed amylase and chitinase activities, which may be relevant in digestive processes.

Still, additional research is required to understand the host-microbiome symbiosis and the microbiome functions provided according to the colonized fraction in shrimp’s digestive tract. Therefore, the present work aimed to examine the microbiota variations along the gastrointestinal tract of the Pacific white shrimp.

Materials & methods

Bioassay: experimental design and sample collection

An initial number of 350 white shrimp (Litopenaeus vannamei) in the postlarvae stage with a mean weight of 0.2 g were obtained from Cruz de Piedra aquaculture farm. After collection, healthy shrimp were acclimatized in a plastic container (450 L) for seven days with aerated filtered seawater (35‰), dissolved oxygen (DO) ≥5 mg/L, temperature 30 °C, water exchange (25%·day−1) and fed daily with a commercial diet.

Shrimp were randomly distributed into fifteen 80 L aquariums incorporated in a recirculation system (RAS). Each aquarium contained 20 shrimp, weighing 0.5 ± 0.1 g. Before shrimp were introduced in the RAS, the experimental units were filled with sterile marine seawater to an operative volume of 60 L. During the bioassay, the RAS’s aquarium conditions maintained a dissolved oxygen (DO) level of 5 mg · L−1, salinity 35 PSU, and temperature 30 °C. These parameters were daily measured using a YSI multiprobe system 556 (YSI Incorporated).

The RAS included a Bubble Bead® Filter (AST Technologies, USA) and two UV sterilizing lamps in a closed circuit to reduce the variability of microenvironmental factors, generating the same constant conditions in all aquariums.

Further, the biofilter containing nitrifying bacteria maintained low concentrations of nitrogen metabolites (0 mg · L−1). Throughout the experiment, which lasted 80 days, shrimp were fed twice a day with commercial feed (4% biomass·day−1) consisting of 25% crude protein, 5% lipids and 4% fiber. Also, uneaten feed, dead organisms and molts were removed daily during the assay. Prior to sampling, shrimp were fastened for 8 h to discard the transitory microbiota. A total of 75 individuals were chosen from the fifteen aquariums, that is, five shrimp per aquarium were randomly selected and pooled. Then, the digestive tract was aseptically dissected and divided into Foregut (stomach), Midgut (hepatopancreas) and Hindgut (intestines). The samples (15 pools) were individually placed in sterile tubes and stored at −80 °C until DNA isolation.

DNA isolation and 16S metagenomic sequencing

Genomic DNA was extracted using a commercial kit (FastDNA Spin Kit for Soil™; MP Biomedicals, Santa Ana, CA, USA) according to the manufacturer’s protocol and mechanically lysed and homogenized using the FastPrep-24™ 5G Instrument (MP Biomedicals™, Santa Ana, CA, USA). The DNA concentration was determined using the Spectra Max fluorescence microplate reader (Molecular Devices, San Jose,CA, USA) and the Quant-iT PicoGreen dsDNA assay kit (Invitrogen Molecular Probes Inc., Eugene, OR, USA) to prepare the amplicon library according to the “16S-metagenomic sequencing library preparation guide” published by Illumina. Finally, the targeted amplification of the 16S rRNA gene V3 region was carried out.

In the case of the Midgut fraction, we only kept five pools (n = 5) from the initially 15 pools because some of the samples showed poor or no amplification. The difficulties of DNA extraction and amplification from the Midgut fraction (hepatopancreas) are related to its high quantity of lipids and enzyme production that made complicated the DNA amplification (Carrillo-Farnés et al., 2007; Cuzon et al., 2004; Schrader et al., 2012). On the other hand, 15 and 14 pools of the Hindgut and Foregut fractions were adequate for the sequencing process.

The library was prepared by performing a PCR using primers flanking the hypervariable V3 region of the bacterial 16S rRNA gene: 338-F (5′-ACTCCTACGGGAGGCAGCAG-3′) and 533-R (5′-TTACCG CGGCTGCTG GCAC-3′) (Huse et al., 2008). All reactions, using 10 ng of DNA template, were adjusted to 20 µl; then these were amplified by PCR (95 °C for 3 min, and 25 cycles at 95 °C for 30 s, 55 °C for 30 s and 72 °C for 30 s, with a final extension at 72 °C for 5 min). A second eight-cycle PCR (95 °C × 30 s; 61 °C × 30 s; 72 °C × 5 min) was performed to add Illumina indexes, using Nextera XT Index Kit (Illumina, San Diego, CA, USA). The PCR amplicons were run through an electrophoresis agarose gel, purified with Ampure XP magnetic beads (Beckman Coulter, USA), and quantified using the Qubit 3.0 fluorometer (Thermo Fisher Scientific, Waltham, MA, USA). Finally, an equimolar pool (4 nM) of the libraries was denatured and diluted to 2 pM. All the pool libraries were loaded and sequenced using an Illumina Miniseq instrument (Illumina Biosystems, San Diego, CA, USA) using standard conditions (300 cycles, 2 × 150). All raw paired-end reads have been deposited in the Sequence Read Archive (SRA) database at NCBI under BioProject ID PRJNA701069.

Bioinformatics analyses

Sequencing reads of the 16S rRNA gene were processed with Quantitative Insights Into Microbial Ecology 2 (QIIME2) pipeline, version 2018.8 (Bolyen et al., 2018). For quality control, demultiplex sequences were denoised, joined in paired reads, and chimeras were removed using DADA2, a plugin of QIIME2 (Callahan et al., 2016). Based on sequence quality plots as a guide, the first two nucleotides were trimmed, and the reads were truncated to 143 bases (Cicala et al., 2020).

After removing low-quality scores, taxonomy was assigned to amplicon sequence variants (ASVs) using a pre-trained classifier Silva_132 with OTUs clustered at 99%. Unassigned sequences and low confidence (0.005%) ASVs were removed, meaning that ASVs with frequency <4 reads were discarded before further analyses (Cornejo-Granados et al., 2018).

A rooted phylogenetic tree was generated for microbial diversity analyses. Multiple sequence alignment of ASV representative sequences was performed using MAFFT software (Katoh & Standley, 2013). Further, FastTree (Price, Dehal & Arkin, 2010) software was used for building a phylogenetic tree, which shows the relationships of different bacterial species in a tree-like model that includes nodes.

Data analysis and statistics

Library sizes were adjusted, rarefying the number of reads to a subsampled minimum depth of 5,098 reads to avoid unequal sample sizes and estimated alpha and beta diversity. A rarefaction curve was generated using ASVs to estimate species richness (alpha diversity) with the qiime diversity alpha-rarefaction plugin implement in QIIME2 (Bolyen et al., 2018). Also, alpha diversity was calculated using Chao1 and Shannon index and Observed ASV among the different digestive tract fractions. The raw count of ASVs and ASVs clustered at the family level were used to calculate the indexes. A Kruskal and Wilcoxon statistic test (p < 0.05) was performed in the raw ASV set. Index comparison at the family level, a one-way ANOVA, and a Tukey statistical test (p < 0.05) was used for the multiple pairwise comparisons among the digestive tract fractions/tissues.

To evaluate the microbiome community structure, the compositional similarity/dissimilarity between samples (beta diversity) was estimated by generating weighted UniFrac, Jaccard and Bray-Curtis dissimilarity matrices. The dissimilarity between fraction samples was visualized using principal coordinate analysis (PCoA) (all listed above performed using QIIME2 diversity plugin). Pairwise comparison of the digestive tract beta diversity distances was performed using permutation multivariate analysis of variance (PERMANOVA) through 4,999 permutations with a p-value of 0.0002 to the beta diversity analysis of QIIME.

To identify the significant taxa in the digestive tract fractions, we also test a Differential Abundance (DA) analysis to identify the significant taxa in the digestive tract fractions. A linear regression framework carried out the multigroup differential abundance analysis with Bias correction for the unobservable differential sampling fractions across samples (i.e. sampling fraction of a sample drawn from a unit volume of an ecosystem) (Lin & Peddada, 2020). According to the fraction co-variability: Midgut (n = 5), Hindgut (n = 15) and Foregut (n = 14), the feature table was agglomerated at the family level. The DA analysis uses the absolute abundance of intersected families’ microbiome data in the digestive tract fractions (76 families, then). Here, we used prevalence to filter the absolute abundance of microbiome data as a form of unsupervised filtration (total abundance > 1%). Since filtering was unsupervised upon the taxonomic annotations, the undetermined families were labeled with their last incomplete lineages. In this case, we assumed that the features (before filtering steps in the bioinformatic workflow, such as chimeric removal, quality read filtering, low filtering features, etcetera) represent a combination of real biological variants that remain insufficiently documented due to a lack of representatives in the reference database (Hibbett & Glotzer, 2011). Finally, an occurrence-based visualization was performed to show the intersection of the number of families through the three digestive tract fractions (Conway, Lex & Gehlenborg, 2017).

In addition to the occurrence-based visualization, we quantified the intra-order interaction of covariates by using non-binary co-expression networks analysis (Weighted topological, wTO). The wTO method measures a link-to-link threshold of significance p adjusted (padj) value <0.01 (p-values, adjusted for multiple testing). Here, positive correlations (blue color links) may represent symbiotic or commensal relationships, while negative correlations (red color links) may represent predator-prey interactions, allelopathy, or competition for limited resources (Gysi et al., 2018).

Functional metagenomic prediction

A PICRUSt2 pipeline was used for functional metagenomic prediction (Douglas et al., 2020). The files of ASVs abundance table and represented sequences were also used as an input to assess functional predictions and matching ASVs normalized by 16S rRNA copy number against the Kyoto Encyclopedia of Genes and Genomes (KEGG) database. The implemented commands were place_seqs.py, hsp.py, and metagenome_pipeline.py to create the predicted metagenomics table. Finally, the Statistical Analysis of Metagenomic Profiles (STAMP) software v2.1.3 was used to analyze the metagenomes predicted by PICRUSt2. Significant differences in the KEGG pathways between the microbiota from the digestive tract fractions were monitored using the non-parametric Kruskal‒Wallis test and Bonferroni approach for multiple test correction, and a p < 0.05 was considered significant.

Results

Sequencing data and ASV classification

A total of 1,609,904 16S rRNA raw reads were generated during sequencing. Quality control analysis returned 1,086,676 sequences (67.49%) that were clustered into 614 ASVs. The rarefaction curves showed that the microbial community was correctly represented by the sequencing data (Fig. S1). Alpha diversity was calculated using Chao1 and Shannon index and Observed ASV among the different digestive tract fractions using two ASVs sets. Regarding the raw ASVs count, no significant differences were detected (p < 0.05). When the ASVs were clustered to the family level, only the Shannon diversity Index showed differences (p < 0.05) between Hindgut-Midgut and Foregut-Midgut comparisons (Fig. 1).

Figure 1 Alpha diversity of the microbiota from the three digestive tract fractions of the white shrimp (Litopenaeus vannamei).

The Chao, Observed, and Shannon indexes were calculated based on different ASVs resolution levels. (A) Indexes were calculated based on the Raw ASVs count. (B) Indexes calculated based on the ASVs clustered at the family level.

In order to test differential groups of taxa, the feature table was agglomerated at the family level, using prevalence filtering as a form of unsupervised filtration. Prevalence analysis reported that Proteobacteria had the highest spread, followed by Actinobacteria, Verrucomicrobia, Firmicutes, Bacteroidetes and Cyanobacteria (Fig. S2). A whole set of 577 features were retained after this filtration step (180,879 [0.996%] absolute abundance). A set of 31 families was intersected through the three fractions; 33, seven, and five intersections were found in Hindgut-Foregut, Midgut-Foregut and Hindgut-Midgut, respectively. A set of families were exclusive per fraction: Midgut (15), Foregut (17) and Hindgut (19). Families presented only in a single fraction were excluded from the DA test (Fig. 2).

Figure 2 The occurrence of families across the digestive tract fractions of the white shrimp (Litopenaeus vannamei).

The numbers at the top of the bars are the size of intersections. The connected dots in the bottom of the bars correspond to the families intersected through the three fractions. The non-connected dots (colored) are the exclusive families in one of the fractions.

Analysis of the taxonomic profile

A barplot was performed with the relative bacterial abundance at the phylum level for each fraction (Fig. 3). The ASV abundance was variable across the three digestive tract fractions. Here, Actinobacteria, Bacteroidetes, Firmicutes and Proteobacteria were represented in the three digestive tract fractions. Particularly, Proteobacteria was the enriched phyla in all the fractions, whereas Actinobacteria was enriched in the Hindgut fraction. Conversely, Verrumicrobia was exclusive in either fractions, Foregut and Hindgut tissues. The results obtained with the DA test showed a similar pattern in the shared and exclusive taxa groups presented in the three digestive tract fractions.

Figure 3 Relative abundance of microbiota enriched in the digestive tract fractions of the white shrimp (Litopenaeus vannamei) at the phylum level.

Abundance <1% was grouped in the low abundance category.

Additionally, a relative abundance bar plot with the DA families (significance < 0.05) across the digestive tract fractions was performed. The bar plot (Fig. 4) shows a high percentage of classified ASVs belonging to the Rhodobacteraceae (74%) and Rubritalaceae (80%) in the Foregut fraction. Also, Cellulomonadaceae (73%) was an enriched family, particularly in the Hindgut. The Aeromonadaceae (90%) and Beijerickiaceae (88%) families registered a great abundance range in the Midgut fraction.

Figure 4 Relative abundance bar plot with the differential abundance (DA) (significance < 0.05) microbial families enriched in the digestive tract fractions of the White shrimp (Litopenaeus vannamei).

Y-axis is ordered as in Fig. 5.

Figure 5 Principal coordinate analysis (PCoA) plot based on weighted UniFrac (A), Jaccard (B), and Bray Curtis distance measurements of beta diversity associated with the gut microbiota of the White shrimp (Litopenaeus vannamei).

The Foregut (stomach) is represented by red color; blue corresponds to the Midgut (hepatopancreas), and yellow belongs to the Hindgut (intestine).

Bacterial community structure of the digestive tract fractions

PCoA analysis revealed significant differences among fractions from the digestive tract (p < 0.05). Figure 5 shows the PCoA of beta diversity associated with microbiota variance for Foregut (stomach), Midgut (hepatopancreas) and Hindgut (intestine), using Weighted UniFrac (A), Jaccard (B) and Bray Curtis (C) distances. Beta diversity metrics were evaluated with PERMANOVA with 4,999 permutations with a p-value of 0.0002. Weighted UniFrac metrics (Fig. 5A) registered 72.67% of the total variance. On the contrary, the principal components of Jaccard (Fig. 5B) represented an accumulated variance of 27.3%, while Bray-Curtis clustering (Fig. 5C) showed the highest value of 53% of the total variability (PC1 24%, PC2 18% and PC3 11%). Despite total variability differences among PCoA, microbial beta-diversity’s community structure using all the metrics presented three primary groups for each organ where Hindgut and Foregut registered a more remarkable similarity.

Differences in the microbial abundance from the digestive tract fractions

The differential abundance (DA) analysis with the analysis composition of microbiomes with bias correction was performed to identify significant family abundance differences among the digestive tract fractions (Fig. 6). Furthermore, the number of families observed in the shrimp Foregut (stomach), Midgut and Hindgut var according to the phylum and fraction. For instance, sixteen significant families were observed from the phylum Proteobacteria. In the case of the phylum Actinobacteria, five significant families were detected. Verrumicrobia registered only one family, Firmicutes three families and Bacteroidetes two families.

Figure 6 Differential abundance (DA) analysis with bias correction of the significant families among the digestive tract fractions.

* Significant at 5% level of significance; ** significant at 1% level of significance; ***significant at 0.1% level of significance.

The DA analysis compares the family abundance between Foregut, Midgut and Hindgut fractions. In this regard, the microbial communities’ abundance varied considerably depending on the digestive tract fraction. In the Foregut fraction, the Rhodobacteraceae and Rubritalaceae families are significantly more abundant than in the Midgut and Hindgut. Otherwise, in the Midgut fraction, three families (Beijerinckiaceae, Aeromonadaceae and Propionibacteriaceae) were significantly higher in terms of abundance than in the other two digestive tract fractions. However, other families registered differences in the Midgut fraction against Hindgut or Foregut. For instance, Micrococcaceae registered a greater abundance in the Midgut, with significant differences only against|the Hindgut. Finally, the Cellulomonadaceae family resulted in a high proportion in the Hindgut, followed by Rhodobacteraceae and Ilumatobacteracea. Notably, only Rhodobacteraceae was significantly superior in the Foregut and Hindgut in comparison with the Midgut fraction.

Bacterial interactions at order level using wTO

In the weighted topological (wTO) network analysis, we found cumulative features at order level, with at least one significant interaction: 37 interactions were found in the Hindgut (44 Orders), 8 interactions in the Midgut (34 Orders), and 34 in Foregut fraction (43 Orders) at 99% confidence (padj-value <0.01). There were five clusters in the Foregut fraction (Fig. 7), although the interactions of cluster 1 and 2 showed great strength, there was greater “shade” (which represent how strong the interaction is) and a more significant contribution in the cluster 5; especially the order Pseudomonadales seems to be the “key” for the connection or communication to the order Bacillales and Lactobacillales. On the other hand, Micrococcales and Cellvibrionales were negatively correlated with Pseudomonadales, Propionibacteriales, Cellvibrionales. The interaction network in the Midgut was poor, maybe related to the sample number (Fig. 8). In this fraction, we found two clusters; cluster 1, showed a network only with the Proteobacteria phylum in equal contribution. On the other hand, Propionibacteriales in cluster 2 was the center of the cluster and the gate of interaction from a more diverse microbiota, such as Desulfobacterales and Cyptophagales. Finally, the Hindgut fraction interactions (Fig. 9) were highly different from the other two fractions. Four clusters were in the Hindgut. Positive interactions were mainly found in all the clusters, except for cluster 4, which suggests a set of bacterial species (Lactobacillales and Vibrionales) that do not coexist with other bacteria from Cluster 2 and 3. The Costridiales order in Cluster 3 contributed with the highest number of microorganisms, as well as the Rhizobiales. These seem to be directly antagonists to the order Vibrionales.

Figure 7 Network interaction graph for microbial communities of the Midgut fraction at the order level, using weighted topological (wTO) network analysis.

Blue color links are positive correlations, while negative correlations are in coral color links. The shade of the links means how strong the interaction is, as well as the degree of interaction within the orders is represented by the node size.

Figure 8 Network interaction graph for microbial communities of the Hindgut at the order level, using weighted topological (wTO) network analysis.

Blue color links are positive correlations, while negative correlations are in red color links. The shade of the links means how strong the interaction is, as well as the degree of interaction within the orders is represented by the node size.

Figure 9 Network interaction graph for the hindgut microbial communities at the order level, using weighted topological (wTO) network analysis.

Blue color links are positive correlations, while negative correlations are in red color links. The shade of the links means how strong the interaction is, as well as the degree of interaction within the orders is represented by the node size.

Predicted metabolic functions of the digestive tract microbiota

Bacterial metabolic functions from digestive tract fractions were enriched and predicted based on KEGG. The Nearest Sequenced Taxon Index (NSTI) was estimated and calculated per sample, and 10 of 900 ASVs resulting above the max NSTI cut-off of 2.0 were removed.

A heatmap was performed of the KEGG predicted pathways across all the digestive tract fractions (Fig. 10). The three principal predicted functions with greater abundance in the digestive fractions studied were related to Metabolism (74%), Genetic Information Processing (14.8%), and Cellular Processes (4.05%). Specifically, in the Foregut, the enriched pathways were related to Metabolism and Cellular Processes. While in the Midgut, besides metabolism, pathways related to genetic information processing and organismal systems were also registered. Likewise, the Hindgut also registered Metabolism and Genetic information processing predicted pathways and Environmental Information Processing. This last pathway was only enriched in this digestive tract fraction.

Figure 10 Heatmap of microbial functions from fractions of the digestive tract of the white shrimp (Litopenaeus vannamei), predicted using level-1 and level-2 of the KEGG database.

Kruskal–Wallis H test was used to analyze the significant differences of species among four groups. KEGG, Kyoto Encyclopedia of Genes and Genomes. Values from the color scale represented the abundance of predicted functions annotated for each KEGG level-1 category.

Also, differentially abundant predicted functional features between the digestive tract fractions were compared and analyzed using STAMP. A total of 27 KEGG pathways with significant differences (p < 0.05) were detected based on the Kruskal–Wallis test. As shown in Table 1, regarding metabolism, the enriched subcategories level-2 pathways in the Foregut were Amino acid metabolism, Carbohydrate Metabolism, Energy metabolism and Xenobiotics biodegradation and metabolism. Notably, Carbohydrate metabolism (C5-Branched dibasic acid metabolism and Glyoxylate and dicarboxylate metabolism) was significantly higher in this digestive tract fraction. The cellular processes predicted pathway was also found in this fraction, with genes involved in the cell growth pathway such as apoptosis. In particular, this level-3 of the functional subcategory was significant in the Foregut.

Table 1 Enriched KEGG pathways (level 1, 2 and 3) according to the digestive tract microbiota of the digestive tract of white shrimp (Litopenaeus vannamei) with significant differences (p < 0.05) based on the Kruskal–Wallis test.

Fraction	KEGG Level-1 and Level-2	KEGG Level-3	
Foregut	Metabolism; Amino acid metabolism	Glycine, serine and threonine metabolism	
Metabolism; Amino acid metabolism	Valine, leucine and isoleucine biosynthesis	
Metabolism; Amino acid metabolism	Arginine and proline metabolism	
Metabolism; Amino acid metabolism	Lysine degradation	
Metabolism; Carbohydrate metabolism	C5-Branched dibasic acid metabolism	
Metabolism; Carbohydrate metabolism	Glyoxylate and dicarboxylate metabolism	
Metabolism; Energy metabolism	Carbon fixation pathways in prokaryotes	
Metabolism; Metabolism of other amino acids	D-Arginine and D-ornithine metabolism	
Metabolism; Xenobiotics biodegradation and metabolism	Chloroalkane and chloroalkene degradation	
Metabolism; Xenobiotics biodegradation and metabolism	Polycyclic aromatic hydrocarbon degradation	
Cellular processes; Cell growth and death	Apostosis	
Ribosome biogenesis in eukaryotes; Translation	Ribosome biogenesis in eukaryotes	
Midgut	Genetic information processing; Folding, sorting and degradation	Protein processing in endoplasmic reticulum	
Genetic Information Processing; Folding, sorting and degradation	Sulfur relay system	
Metabolism; Amino acid metabolism	Tryptophan metabolism	
Metabolism; Lipid metabolism	Glycerophospholipid metabolism	
Metabolism; Metabolism of cofactors and vitamins	Folate biosynthesis	
Metabolism; Metabolism of cofactors and vitamins	Vitamin B6 metabolism	
Metabolism; Metabolism of terpenoids and polyketides	Carotenoid biosynthesis	
Organismal systems; Endocrime system	Insulin signaling pathway	
Metabolism; Xenobiotics biodegradation and metabolism	Toluene degradation	
Hindgut	Metabolism; Metabolism of cofactors and vitamins	Nicotinate and nicotinamide metabolism	
Environmental information processing; Membrane transport	Phosphotransferase system (PTS)	
Metabolism; Energy metabolism	Nitrogen metabolism	
Metabolism; Metabolism of cofactors and vitamins	Ubiquinone and other terpenoid-quinone biosynthesis	
Metabolism; Xenobiotics biodegradation and metabolism	Drug metabolism-other enzyme	
Genetic information processing; Replication and repair	DNA replication	

Likewise, in the Midgut fraction, the enriched metabolism pathways included Amino acid metabolism (Tryptophan metabolism), Metabolism of cofactors and vitamins (Folate biosynthesis and Vitamin B6 metabolism), Metabolism of terpenoids and polyketides (Carotenoid biosynthesis), Xenobiotics biodegradation and metabolism (Toluene degradation), and Lipid metabolism. Interestingly, bacterial communities from the Midgut fraction had a significantly greater abundance of Lipid metabolism, with the level-3 subcategory glycerophospholipid metabolism pathway and metabolism of terpenoids and polyketides such as carotenoid biosynthesis.

In contrast, five predicted pathways were enriched in the Hindgut fraction. Metabolism of cofactors and vitamins (Nicotinate and nicotinamide metabolism), Energy metabolism (Nitrogen metabolism) and Xenobiotics biodegradation and metabolism were the metabolic pathways enriched. Other enriched functions included Environmental information processing such as Membrane transport and metabolism (Phosphotransferase system) and Genetic information processing that includes the Replication and repair pathway.

Distances of the predicted bacterial functions among the digestive tract fractions in a principal component analysis (PCoA)

A principal component analysis (PCoA) was performed on the relative abundance values of the KEGG pathways of the digestive tract microbiota. The PCoA showed a clear distinction between the clustering on the predicted functions from the microbiota of the digestive tract’s three fractions. The predicted functional profile showed significant differences among the fractions, based on the Kruskal–Wallis test and the Bonferroni approach for multiple test correction using STAMP. Despite the significant differences, Midgut predicted functions seemed to be apart from the Foregut and shared few functions with the intestinal microbiota (Fig. S3).

Discussion

The shrimp’s gut microbial ecosystem is a whole essential system playing vital symbiotic roles in maintaining homeostatic physiology in the host. However, most studies about crustacean microbiota have been exclusively focused on the intestine (Cornejo-Granados et al., 2018; Cheung et al., 2015). This study provides an approach to the entire digestive tract, including the Foregut (stomach), Midgut (hepatopancreas) and Hindgut (intestine) fractions.

The relative abundance bar plot with the DA families revealed that, at the family level, Rhodobacteracea was well represented in the Foregut, while in the Midgut was underrepresented. Liu et al. (2019) reported that Rhodobacteraceae could be potentially applied to reduce the influence of cold stress in white shrimp; however, the members of Rhodobacteraceae have diverse physiological and metabolic characteristics and are often found in marine animals as symbiotic bacteria (Pujalte et al., 2014). Besides, Rhodobacteraceae bacteria seem to exhibit probiotic potential due to their ability to produce tropodithietic acid (TDA) to inhibit pathogens. These can also synthesize vitamin B12, which is essential for shrimp growth (Sañudo-Wilhelmy et al., 2014). Notably, this family was mainly found in the Foregut fraction. This fraction consists of a stomach connected to an esophagus with further divisions (cardiac chamber and pyloric chamber) (Muhammad et al., 2012). Mouthparts perform food particles’ intake; food passes the esophagus and reaches the stomach for mechanical and extracellular digestion (McGaw & Curtis, 2013). The high abundance of Rhodobacteraceae could be associated with its action against pathogenic bacteria as a first defense barrier at the beginning of the digestive tract required by the host (Soonthornchai et al., 2015).

On the other hand, the Cellulomonadaceae family was mainly found in the shrimp Hindgut, and this family has been reported in shrimp ponds (Wang et al., 2017a). A large variety of hydrolytic starch, xylan and cellulose-degrading enzymes, are produced by Cellulomonadaceae, which belongs to the Actinobacteria phylum. Short-chain fatty acids, such as acetate, are produced by some Actinobacteria that promote the host epithelial cells’ defense functions and protect the host from lethal infections (Fukuda et al., 2012). Specifically, genera from this family, like Actinotalea can degrade cellulose and produce acetate (Stackebrandt & Schumann, 2014), which has high inhibitory activity against pathogenic Vibrio species in L. vannamei (Da Silva et al., 2013).

The family Beijerinckiaceae was only detected in the Midgut (hepatopancreas). It belongs to the order Rhizobiales within the phylum Alphaproteobacteria; the main genera represented in this family was Methlylobacterium. This group encompasses aerobic bacteria capable of forming poly-β-hydroxybutyrate granules and fixing nitrogen; furthermore, they can increase exopolysaccharide production (Marín & Arahal, 2014). Fixing nitrogen bacteria in aquatic organisms as symbionts have been poorly studied; however, nitrogen fixation processes within the microbial community from the digestive tract of teleost species, like the catfish, Panaque nigrolineatus, have been reported (McDonald et al., 2015). According to previous reports performed by Lu et al. (2016), there are genes and pathways involved in nitrogen metabolism that play an essential role in reducing ammonia toxicity in the hepatopancreas of L. vannamei; also, ammonia stress could inhibit the immune system and increase the susceptibility of shrimp to pathogens. Overall, it seems that the Midgut gland provides a suitable niche for nitrogen fixation that may facilitate the production of reduced nitrogen by the Beijeranckiaceae family.

The microbial communities from the digestive tract’s fractions showed that shrimp gut microbiota varies depending on the fraction sampled. Comparisons of beta diversity Bray-Curtis distances showed consistent separation between the Foregut and Hindgut against the Midgut gland, suggesting that the white shrimp could provide a unique ecological niche according to their digestive tract fraction.

Nevertheless, there were families shared between the digestive tract fractions. Families across the digestive tract showed a set of 31 groups through the three digestive fractions. However, the analysis exhibited that the Foregut and Hindgut were the fractions sharing a higher number of families (33), unlike Midgut-Foregut (seven) and Hindgut-Midgut (five). The more significant number of families shared among Foregut and Hindgut is possibly due to the niche structure’s similarity. Cheung et al. (2015) reported no apparent differentiation between bacterial communities in the Foregut and the Hindgut of the crustacean Neocaridina denticulata because both present a chitinous lining despite the distinct physiological functions of these organs. However, the attachment of bacteria could also be associated with the host’s functional requirements in the Midgut fraction. For example, in other crustaceans, endosymbiotic bacteria contributing to digestive processes thrive in this digestive tract fraction (Zimmer et al., 2001). However, other omics approaches are required to understand how these differences between the gastrointestinal fractions contribute to the symbiotic relationship.

Highly complex interactions were observed in the gut microbiota fractions. In the Hindgut, the gut network was mainly focused on the order Costridiales and Rhizobiales, which could strongly inhibit Vibrionales. A possible explanation of this networking is that these could compete for nutrient resources or adhesion sites, displacing the order Vibrionales. Besides, the order Costridales is a Gram-positive butyric acid-producing probiotic, Clostridium butyricum, which is also described as part of the normal intestine microbiota of shrimp and has a beneficial effect on the intestine health of L. vannamei (Duan et al., 2018). Also, the order Rhizobiales has already been reported to exert an antagonistic effect against Vibrionales in postlarval shrimp (L. vannamei) (Cao et al., 2020). Some vibrios (as Vibrionales) are recognized as opportunistic pathogens causing vibriosis in shrimp (Chandrakala & Priya, 2017); however, the above antagonistic network (and others still unknown) could explain the presence of pathogenic strains without producing harmful effects in shrimp.

Along with the taxonomic profile, this approach provides insight into the gut microbiota’s functions. In the long-term, the predicted functional profiles from the digestive tract fractions exhibited differences. We found that the digestive tract microbiota could play a key role in host nutrition. In this study, the differences in predictive functional pathways between the digestive tract fractions were based on their microbial composition and functional capabilities. Our results indicate that genes involved in Carbohydrate Metabolism showed greater abundance in the Foregut fraction. In this regard, shrimp has a limited capacity to metabolize nutrients like carbohydrates; alternatively, shrimp uses proteins as a source of energy and growth (Rosas et al., 2000). However, the harbored microbiota presents Carbohydrate Metabolism (Glyoxylate and dicarboxylate metabolism), probably contributing with metabolic power to cope with the host’s deficiencies. Ma et al. (2020) reported a relationship between carbohydrate metabolism pathways and the Vibrio harveyi infection’s progress due to the high energy demand for shrimp survival (Wang et al., 2017b).

The hepatopancreas is the main secretory organ of shrimp and the primary absorption barrier of crustaceans’ nutrients (Jobling, 2012). Besides, it is the main organ of lipid metabolism. In this regard, the Lipid metabolism (Glycerophospholipid metabolism) pathway and the metabolism of terpenoids and polyketides such as carotenoid biosynthesis were representatives of the Midgut fraction. Enzymatic activity from shrimp digestive tract microbiota has already been reported; specifically, Vibrio and Pseudoalteromonas strains isolated from hepatopancreas, presented amylase, chitinase, lipase and esterase activities, indicating that L. vannamei microbiota includes some groups exhibiting multi enzymatic activity that contributes to the degradation of dietary components (Tzuc et al., 2014).

Also, the hepatopancreas is an important indicator of crustaceans’ host health condition (Deng et al., 2017). Previous studies of the crayfish’s (Procambarus clarkia) hepatopancreatic microbiota revealed that diseased organisms underwent dysbiosis in the gut-hepatopancreatic microbiota, affecting lipid synthesis pathways (Wu et al., 2021). Although, the relationship between gut dysbiosis and physiological effects in lipid synthesis remains unknown.

Lipids are essential in aquatic animals to reduce osmotic shock due to preserving the ion balance and regulating biological membranes’ structure. Glycerophospholipids, glycerol-based phospholipids, are the main biological membranes component (Chen et al., 2014; Palacios et al., 2004). In the study reported by Chen et al. (2015) lipid metabolism is involved in osmoregulation strategies in essential organs in the white shrimp.

Not only the Lipid metabolism pathway was enriched in the Midgut but also Carotenoid biosynthesis. Carotenoids are lipid-soluble tetraterpenoid compounds, and, in general, animals are cannot biosynthesize these compounds. Plants and microorganisms like bacteria primarily biosynthesize these. However, carotenoids are essential antioxidants and play crucial functions in health (Tanumihardjo, 2012). Additionally, they are natural pigments involved in animals’ body coloration; for instance, astaxanthin is a carotenoid largely responsible for the color of shrimp, and diets for shrimp aquaculture are added with this carotenoid to improve its visual appeal (Su, Huang & Liu, 2018).

Pathways related to the metabolism of cofactors and vitamins (Nicotinate and nicotinamide metabolism) were more abundant in the Hindgut microbiota than other digestive segments. Previously there are reports of microbial sequences annotated to energy metabolism, xenobiotics biodegradation, membrane transport in intestinal microbiota from two cultured stages (30 d and 60 d) of L. vannamei (Gao et al., 2019), even so, their implications in the intestine shrimp are not yet elucidated. However, regarding energy metabolism, such as nitrogen metabolism, Ortiz-Estrada et al. (2021) reported genes encoding enzymes implicated in the nitrogen metabolism in probiotic biofilm from a zero recharge shrimp culture, which registered better water quality parameters than a heterotrophic biofilm. Nitrogenous metabolites, particularly unionized ammonia (NH3) and nitrite (NO2) are toxic for shrimp. Consequently, the ideal condition requires the rapid oxidation of ammonia/ammonium to nitrite and, then, to nitrate. Such processes are carried out in aerobic environments by ammonia-oxidizing bacteria and nitrite-oxidizing bacteria (Robles-Porchas et al., 2020). On the whole, metabolic predicted functions in the digestive tract fractions are necessary for adequate host development.

Finally, the taxonomic and predicted functional differences among gut fractions suggest that these could provide specialized functions for each organ. As stated above, the association of diseases with dysbiosis in organs, like the intestine and the hepatopancreas in some crustaceans, indicates that the microbiota can play a protective role against pathogens in the different gut fractions. However, these microbiotas’ communication with the crustacean systems is still hypothetical and should be addressed in further approaches.

Conclusions

The three different digestive tract compartments showed that bacterial populations varied substantially between the shrimp gut regions, especially between the Midgut (hepatopancreas) and Hindgut (intestine). The Midgut (hepatopancreas) seems to be a fraction harboring exclusive microbiota since it was the tissue with less diversity and less vulnerable to its microbiota changes. Not only the microbiota shifted in accordance with the digestive tract component, but also their predicted functions. In the Foregut, carbohydrate metabolism was mainly registered in this fraction. On the contrary, pathways related to lipid metabolism prevailed in the Midgut. In the Hindgut, pathways like the metabolism of cofactors and vitamins together with energy metabolism were enriched in this fraction. Finally, lipid metabolism pathways seem to be exclusively and essential in the Midgut. According to host physiology, the present study contributed to understanding the differences in microbiota composition harbored by the different fractions of shrimp gut, providing a basis for studying the microbiota’s role in the different gut fractions.

Supplemental Information

Supplemental Information 1 Rarefaction curves constructed based on observed ASVs.

Click here for additional data file.

Supplemental Information 2 The Fraction of Prevalence (y axis) versus Total abundance (x axis) shows a positive relationship.

Proteobacteria (0.643), Actinobacteria (0.743), Verrucomicrobia (0.767), Firmicutes (0.614), Bacteroidetes (0.831), Cyanobacteria (0.983). Low taxa are the group of taxa with an abundance lower than 1% of the total abundance.

Click here for additional data file.

Supplemental Information 3 Principal component analysis (PCA) profile of predicted functions from digestive tract microbiota of white shrimp (Litopenaeus vannamei).

The percentage of variation explained by PC1, PC2 and PC3 are indicated in the axis. Different colors represent different samples. Comparison of predicted microbial function among groups based on KEGG level-3.

Click here for additional data file.

Additional Information and Declarations

Competing Interests

Author Contributions

Data Availability

The authors declare that they have no competing interests.

Estefanía Garibay-Valdez conceived and designed the experiments, performed the experiments, analyzed the data, prepared figures and/or tables, authored or reviewed drafts of the paper, and approved the final draft.

Francesco Cicala performed the experiments, analyzed the data, prepared figures and/or tables, and approved the final draft.

Marcel Martinez-Porchas conceived and designed the experiments, performed the experiments, analyzed the data, prepared figures and/or tables, authored or reviewed drafts of the paper, and approved the final draft.

Ricardo Gómez-Reyes analyzed the data, prepared figures and/or tables, and approved the final draft.

Francisco Vargas-Albores performed the experiments, analyzed the data, authored or reviewed drafts of the paper, and approved the final draft.

Teresa Gollas-Galván performed the experiments, analyzed the data, authored or reviewed drafts of the paper, and approved the final draft.

Luis Rafael Martínez-Córdova performed the experiments, authored or reviewed drafts of the paper, obtained live organisms from commercial farms, and approved the final draft.

Kadiya Calderón performed the experiments, analyzed the data, prepared figures and/or tables, authored or reviewed drafts of the paper, and approved the final draft.

The following information was supplied regarding data availability:

All raw paired-end reads are available in the Sequence Read Archive (SRA) database at NCBI under BioProject ID PRJNA701069.

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
