# Peer review of "Longitudinal variations in the gastrointestinal microbiome of the white shrimp, Litopenaeus vannamei"

_PeerJ, doi:10.7717/peerj.11827_

## Round 0.1 · original submission · Major Revisions

These days metagenomics studies are getting more important due to their robustness in identifying bacterial communities and the functional annotation as well. The present work is also very interesting but the second reviewer raises some basic queries that you must address properly. Please address all other queries of honorable reviewers in a revised submission.

Reviewer 1 ·

Basic reporting

The manuscript does a reasonably good job at describing the microbial communities in white shrimp but it significantly lacks adequate interpretation of the results. This is especially evident in the underannotated figures (i.e. what do the values mean in Fig. 2; what are the values in the heatmap in Fig. 6). Additionally, there's several minor grammatical errors that make comprehending the narrative difficult.

Experimental design

The manuscript fails to adequately describe the experimental design (i.e. L107-L118, how many samples were included in the analyses; were the samples comprised of pools of shrimp gut tissues or individual tissues), making it difficult to understand how well the observed results can be generalized across white shrimp. This should be made much clearer.

Validity of the findings

The findings, while lacking in interpretation, seem fine, but this is somewhat hard to assess given that it is unclear what the dataset looks like.

·

Basic reporting

The paper entitled: “Microbiota variations through the gastrointestinal tract of the white shrimp, Litopenaeus vannamei” by Garibay-Valdez et al, reports the microbiome composition of three portions of the GI tract in the white shrimp.
I think the study is potentially interesting, but needs improving before publication. There are some minor errors in the English language that need to be corrected (there is an abuse of the possessive case for instance). Moreover, some points of the paper are unclear to me.
I think the aim of the study should be better explained: why is it so important to look at the 3 portions, what is the “catch”. You report the economic value of the shrimp, but is the different localization of the bacterial species changing anything in the strategy of aquaculture for example? Or is it just interesting for the scientific point of view? If so you should mention.
I think the results is missing a summary of the composition of the samples, at different taxa levels, what are the main phyla, classes…etc? can you represent this in a barplot? I think it would be nice to see it graphically to highlight any difference among the 3 sampling sites.
In general, I think some parts of the manuscript would be better presented in a table, ie the Pathways in line 300-331. I see you refer to tab1 but there is no mention in the tables that these Pathways are the one overrepresented.

Experimental design

First of all, how did you sample the 3 portions of the GI? Did you sample them from the same animals? I think the best way to account for individual variability would be to sample the 3 portions from all the animals. You did not justify why you have a different number of samples for the 3 portions. I could not see any reference to the final number of samples the authors studied. Moreover, why did you divide in groups? How many animals/group? What are the differences among the groups?
Finally. The results of alpha and beta-diversity need to be improved: is there a difference in observed ASV number and/or Shannon index in the 3 portion of the gut? What are the R2 and P values for the 3 distances you calculated?
In general, why did you decide to focus on the families only? What about at higher taxonomic levels?
Did you sample the water? It would be great to have that info to see if some of the species are found there.

Validity of the findings

I cannot comment on this before I understand how many samples where included in the study.
I think the novelty of the study is well stated, but I do not get the importance of studying the difference localization of the bacterial community.

Additional comments

please refer to my previous comments to implement the paper. Find here also some more detailed comments:

Title: I do suggest to use “Longitudinal variations of the microbiome of the gastrointestinal tract of the white shrimp, Litopenaeus vannamei”
Line 28: of the foregut
Line 30 and 37: I would not use “potential”
Line 42-43: relies on the fraction… is not clear, please rephrase.
Line 44-45: providing… is not clear, please explain.
Line 49-50: I would report here the main diseases that interest this species.
Line 58: ‘Provides beneficial host functions…” is confusing. Please rephrase
Line 59-62: this sentence is missing the verb
Line 70: maybe use “digestive”
Line 75: this sentence is confusing
Line 98: at this point the authors should state the number of animals
Line 107: what are the differences among the groups?
Line 107-8: I am confused on the number of shrimps: 5 groups *3 replicates*20 animals?
Line 111: use always the same units (you used before mg/Ml at lune 101, then you used the “-1’.
Line 113-114: what is “almost 0”?
Line 131: please report the reference for the primers
Line 140: past of load is loaded
Line 148: denoised
Line 165: did you calculate any alpha-diversity indexes?
Line 179: are these the total numbers of
Line 180 and 215: I would use clustered or parsed?
Fig1: I am not sure what the graph represents
Fig3 is cut in two points
Line 235: it is ASVs not ASV’s
Fig4: there is a typo in the caption. How many animals you had? How you did not have the same number for the 3 locations? Did you extract the DNA from the same samples? If you did not, how can you be sure the differences are not due to the individual variability?
Line 245: can you provide the results of the beta-diversity tests? Including R2 and P values
Line 257: use the past tense and the same all along the manuscript
Line 278: is this order in the “taxonomic group” sense?
The figures are not in the right order.
Line 301-2: is this true for the 3 locations?
Line 308: fractions
Fig7 can go on the supplementary material
Why didn’t you put a Ven diagram to report the shared taxa? And why did you chose to report only family level?
Line 433: and in italics

Reviewer 3 ·

Basic reporting

.

Experimental design

.

Validity of the findings

.

Additional comments

I’ve read and analized the work carefully.
It is an original work, in which various tools where used, including molecular biology like bioinformatics to breakdown since the holobionte point of view to the white shrimp Litopenaeus vannamei. Comparing the three most important components in the digestic tract: the foregut (stomach), midgut (hepatopancreas), and hindgut (intestine). With the goal of knowing the genetic structure of the bacterial community using 165 rRNA.
To contribute in the understanding of the functions of said bacterial communities.
The manuscript is clear, professional English language used throughout. I just have minor suggestions for page 6 line 38 it says while, I suggest while.
Lines 40 repeats “and” twice.
Line 257 says “vary” I suggest “varies”
Lines 484 also repeats “and” twice.
The results are fresh, new, and they contribute to the knowledge of the microbiota paper. The structure is clear, the Figures are relevant, well labelled & described.
Rigorous investigation performed to a high technical.
The results are clearly described, for disscusion there is an extensive bibliographic revision which is actualized and opportune.
All of the quotes are on the literature and the other way around.

---

## Round 0.2 · Minor Revisions

Dear Authors, there are very few queries by the reviewers; please address them carefully and resubmit for the acceptance!

Reviewer 1 ·

Basic reporting

The 'new' figure legend for figure 6 does not correspond with what is presented in the figure.
Otherwise, the revisions made, especially those highlighted by the co-reviewer, improve the manuscript.

Experimental design

sufficiently improved.

Validity of the findings

Reasonable.

·

Basic reporting

I was very happy to revise the paper again and notice how it improved after the first round of revisions.
Minor comments:

Unit are still written in different ways: mg/ml or ml-1 or day-1, also check ml or mL
Line 117: shrimps
Line 144: we only kept
Line 195 and 275: 4,999
Line 243: fig s2 csn go to main text
Fig 3 is still cut in some points: ie close to rubritaleaceae

Experimental design

the experimental design is now fully explained in the revised paper.

Validity of the findings

no comment

Additional comments

I very much appreciated the effort the authors put in the revision.

---

## Round 0.3 · accepted · Accept

As white shrimp is one of the major aquaculture species worldwide, it is very important to know their gastrointestinal microbiome for better health management. Thank you for your contribution!